# OpenReview forum: "Reading Books is Great, But Not if You Are Driving! Visually Grounded Reasoning about Defeasible Commonsense Norms"
_EMNLP/2023/Conference — EMNLP 2023 Main_

### Official Review · Reviewer_aeyz · 2023-08-04

**Soundness:** 4

**Excitement:**

4: Strong: This paper deepens the understanding of some phenomenon or lowers the barriers to an existing research direction.

**Paper Topic And Main Contributions:**

The paper construct a new multimodal benchmark for moral detection and judgment. It provides a detailed pipeline of how to construct the dataset,  includes using image descriptions to generate moral text, linking with images, and then manually annotating the data. The method is reasonable and efficient. On the basis, the paper proposes two tasks, experimental results indicate that the two tasks are still challenging for most of current LLMs. At last, the paper presents a new approach to better align models with humans. The paper can be meaningful for the research area of LLMs and moral computing.

**Reasons To Accept:**

The construction method of the dataset is reasonable.
The experimental results are rich and a detailed analysis has been conducted on the performance of LLM.

**Reasons To Reject:**

There is no comparison with traditional ML methods like BERT or T5. How does traditional machine learning perform on this issue? I think it should be reported.

**Reproducibility:**

4: Could mostly reproduce the results, but there may be some variation because of sample variance or minor variations in their interpretation of the protocol or method.

**Reviewer Confidence:**

5: Positive that my evaluation is correct. I read the paper very carefully and I am very familiar with related work.

---

> ### Author Rebuttal · Authors · 2023-08-28
>
> We thank the reviewer for recognizing the soundness of our data construction pipeline. Furthermore, we are pleased that the reviewer acknowledged the richness of our experimental results and the detailed analysis we conducted on the performance of the LLM. Please find a response to the specific question below.
>
> **How does traditional machine learning (like BERT or T5) perform on the task?**
>
> In response to the reviewer's suggestion, we conducted further experiments to evaluate the efficacy of traditional machine learning models, specifically BERT and T5. Given that these models lack zero-shot or few-shot capabilities, we adopted the fine-tuning approach detailed in Section 5 and tested them on the NormLens benchmark. We incorporated image descriptions as additional textual input since these models cannot process visual data. For BERT, we added a linear classifier head, the only trainable module. The results are presented in the table below:
>
> | HA (precision) | Wrong. | Okay. | Impossible. | Avg. |
> |----------------|--------|-------|-------------|------|
> | BERT           | 17.1   | 68.9  | 21.4        | 35.8 |
> | T5 small       | 21.4   | 23.4  | 66.2        | 37.0 |
> | T5 base        | 31.0   | 77.7  | 22.7        | 43.8 |
>
> | MA (precision)    | Wrong. or Impossible. | Wrong. or Okay. | Okay. or Impossible. | Avg. |
> |----------------|--------|-------|-------------|------|
> | BERT           | 31.6   | 82.0  | 90.7        | 68.1 |
> | T5 small       | 83.2   | 22.4  | 87.5        | 64.4 |
> | T5 base        | 46.2   | 78.9  | 96.5        | 73.9 |
>
> From these results, we can draw two primary observations:
>
> - The fine-tuned BERT and T5 models align better with human judgments than some four-shot VLM baselines, although they fall short of the performance when compared with fine-tuned VLM baselines.
> - There appears to be a trend where models with a larger number of parameters perform better on the benchmark --- T5-small (60M params) ~= BERT-base (100M params) < T5-base (200M params).

---

### Official Review · Reviewer_wNre · 2023-08-04

**Typos Grammar Style And Presentation Improvements:** Well written with very minor typos
**Soundness:** 3

**Excitement:**

3: Ambivalent: It has merits (e.g., it reports state-of-the-art results, the idea is nice), but there are key weaknesses (e.g., it describes incremental work), and it can significantly benefit from another round of revision. However, I won't object to accepting it if my co-reviewers champion it.

**Paper Topic And Main Contributions:**

The paper proposes and describes a dataset called NORMLENS for visually grounded reasoning for common sense norms. The common sense assertions / norms are predicated on a context which make the norms right (morally okay), wrong (morally incorrect) or impossible. The paper also describes different architectures (using VLM, SM and purely LLM based) to arrive at a decision and explanation for the decision and alignment with human judgement.

**Reasons To Accept:**

1) A new multi-modal benchmark for testing visually grounded common sense norms that are defeasible.
2) Study of comparative performance of a set of architecture based on Textual Language models, Socratic models (VLM + LM) and VLM both on classification of judgement and power of explain ability
3) Multimodal grounded reasoning data generation pipeline with set of tuning and filtration processes and two tests on alignments (High and Medium alignment)


**Reasons To Reject:**

1) Not enough discussion on the results. It appeared the SM with a VLM model with GPT4 can outperform others. This may be due to GPT4 inherent capability of text grounded reasoning if vision alignment is good enough.

**Reproducibility:**

4: Could mostly reproduce the results, but there may be some variation because of sample variance or minor variations in their interpretation of the protocol or method.

**Reviewer Confidence:**

3: Pretty sure, but there's a chance I missed something. Although I have a good feel for this area in general, I did not carefully check the paper's details, e.g., the math, experimental design, or novelty.

---

> ### Author Rebuttal · Authors · 2023-08-28
>
> We thank the reviewer for recognizing the novelty of our multi-modal benchmark and the value it offers for testing defeasible visually grounded commonsense norms. We further appreciate the reviewer for finding the study of various models on commonsense norm tasks to be valuable, and finding our data generation pipeline to be meaningful. Please see our response to the specific question below.
>
> ### More discussions about the results
>
> Thank you for encouraging us to add more in-depth discussion about the results! We outline some critical discussion points below. We will add the new experimental results and these new discussions to the final paper!
>
> **Capabilities of GPT-4**
>
> As outlined in lines L414-L437 of the main paper, we postulate that the high alignment and explanation scores observed in SM equipped with GPT-4 can be attributed to its innate reasoning capabilities. However, a deep dive into the underlying reasons for its remarkable performance remains elusive due to limited detailed information on GPT-4 (OpenAI et al., 2023). Despite this, we are keen to further explore this avenue and seek a more comprehensive understanding of its proficiency in handling commonsense norms.
>
> While the following two points might not directly address the reviewer’s specific concerns, we've included them to foster a more comprehensive discussion regarding our paper.
>
> **Baselines added for the experiments**
>
> We conducted additional experiments with BERT/T5 models upon the request of reviewer “aeyz”. Given that these models lack zero-shot or few-shot capabilities, we adopted the fine-tuning approach (which is detailed in Section 5 in the main paper) and tested them on the NormLens benchmark. We incorporated image descriptions as additional textual input since these models cannot process visual data. For BERT, we added a linear classifier head, the only trainable module. The results are presented in the table below:
>
> | HA (precision) | Wrong. | Okay. | Impossible. | Avg. |
> |----------------|--------|-------|-------------|------|
> | BERT           | 17.1   | 68.9  | 21.4        | 35.8 |
> | T5 small       | 21.4   | 23.4  | 66.2        | 37.0 |
> | T5 base        | 31.0   | 77.7  | 22.7        | 43.8 |
>
> | MA (precision)    | Wrong. or Impossible. | Wrong. or Okay. | Okay. or Impossible. | Avg. |
> |----------------|--------|-------|-------------|------|
> | BERT           | 31.6   | 82.0  | 90.7        | 68.1 |
> | T5 small       | 83.2   | 22.4  | 87.5        | 64.4 |
> | T5 base        | 46.2   | 78.9  | 96.5        | 73.9 |
>
> From these results, we can draw two primary observations:
>
> - The fine-tuned BERT and T5 models are more aligned with human judgments than some four-shot VLM baselines, although they fall short of the performance when compared with fine-tuned VLM baselines. This result suggests that we need more effort to improve the few-shot/zero-shot capabilities of recent VLM models on defeasible commonsense norms.
>
> - There appears to be a trend where models with a larger number of parameters perform better on the benchmark — T5-small (60M params) ~= BERT-base (100M params) < T5-base (200M params).
>
> **Is visual grounding in the training stage beneficial for the task?**
>
> Answer: Not really. To probe the roots of GPT-4's innate reasoning capabilities, we run additional experiments to investigate the role of visual grounding during the training phase. One hypothesized reason is the assimilation of representations through visual grounding during training.
>
> Given the ambiguity surrounding GPT-4, we turned to LLaVA – a model boasting GPT-4-level visual alignment capabilities – to demonstrate the effects of vision alignment. LLaVA integrates a CLIP visual encoder, a projection layer for visual alignment, and a language model (Vicuna). Throughout its training phase, LLaVA concurrently updates both the pretrained weights of the projection layer and the language model. For our experiment, we extracted Vicuna from the pretrained LLaVA weights and compared it against the standalone Vicuna, under SM conditions. The outcomes are described below:
>
>
> | HA (precision)    | Wrong. | Okay. | Impossible. | Avg. |
> |-------------------|--------|-------|-------------|------|
> | LM (Vicuna)       | 19.8   | 97.7  | 2.3         | 39.9 |
> | LM (LLaVA-Vicuna) | 14.4   | 94.9  | 11.6        | 40.3 |
> | SM (Vicuna)       | 18.7   | 99.1  | 8.3         | 42.1 |
> | SM (LLaVA-Vicuna) | 12.3   | 98.0  | 14.4        | 41.6 |
>
> | MA (precision)    | Wrong. or Impossible. | Wrong. or Okay. | Okay. or Impossible. | Avg. |
> |-------------------|-----------------------|-----------------|----------------------|------|
> | LM (Vicuna)       | 11.7                  | 97.8            | 98.9                 | 69.5 |
> | LM (LLaVA-Vicuna) | 21.9                  | 89.1            | 99.7                 | 70.3 |
> | SM (Vicuna)       | 11.1                  | 98.8            | 100.0                | 70.0 |
> | SM (LLaVA-Vicuna) | 21.9                  | 91.6            | 99.2                 | 70.9 |
>
> From this data, it's evident that the Vicuna extracted from LLaVA doesn't exhibit a marked improvement over the original Vicuna. This insinuates that the role of vision alignment during training has minimal bearing on our benchmark in the SM context.
>
> **References**
>
> - Liu et al., 2023, Visual instruction tuning
> - OpenAI et al., 2023, Gpt-4 technical report

---

### Official Review · Reviewer_SVtk · 2023-08-06

**Soundness:** 4

**Excitement:**

4: Strong: This paper deepens the understanding of some phenomenon or lowers the barriers to an existing research direction.

**Paper Topic And Main Contributions:**

This paper proposes a multimodal benchmark which focuses on visually grounded commonsense reasoning (NormLens). The goals of this benchmark are to test models’ judgment and their explanations behind the judgment. To this end, the task of NormLens is formulated as a combination of three-way classification (OK, Wrong, or Impossible) and explanation generation. Concretely, each example (a multimodal situation) consists of an image (e.g., a person is driving a car.) and a text-based situation (e.g., Is “reading a book” in the context of a given image morally okay?). The labels are a human judgment (OK, Wrong, or Impossible) and a textual explanation. Each multimodal situation can have multiple labels with an average of 4.3 per situation (i.e., judgments from different annotators). Moreover, the multimodal situations are divided into two groups based on the level of annotator agreement. The NormLens dataset includes 2k situations in total.

This paper provides baseline results on NormLens using SOTA unimodal and multimodal models (e.g., GPT4, BLIP2).  LLMs are used as text-only baselines, and only the judgment precision is reported. Socratic models (LLMs + text descriptions of images by a VLM) and VLM (e.g., LLaVa, BLIP2) are also compared. In summary, the Socratic model with GPT4 achieves the best performance both on the judgment precision and explanation generation (measured by BLEU, ROUGE-L, and METEOR), showing the strong results on all three classes. The error analysis shows that many SMs and VLMs struggle with Wrong and Impossible classes. The authors additionally show that extra training data generated by ChatGPT improves the judgment precision of some SMs and VLMs.


**Questions For The Authors:**

- Have you tried finetuning any VLM (e.g., BLIP2) on NormLens (e.g., use MA as training and eval on HA)? I wonder if VLMs can generate good explanations without finetuning.
- What about image diversity? Would you be able to perform an analysis on the images, grouping them based on similarity scores?

**Reasons To Accept:**

- This dataset is a good testbed for the commonsense reasoning abilities of black-box LLMs/VLMs.
- This dataset focuses on social norms/morals which are not heavily covered by prior work (e.g, VCR).
- The data construction process and quality control stages look reasonable.
- This paper is well written and easy to follow.


**Reasons To Reject:**

- Although the situations are checked by human annotators, the seed situations are generated by ChatGPT. The coverage of situation types might be limited.
- The types of situations/social norms (e.g., physical/psychological safety) are not clear in the main paper.
- It’s a bit hard to interpret precision on NormLens-MA, where the different labels could be considered as gold.


**Reproducibility:**

4: Could mostly reproduce the results, but there may be some variation because of sample variance or minor variations in their interpretation of the protocol or method.

**Reviewer Confidence:**

4: Quite sure. I tried to check the important points carefully. It's unlikely, though conceivable, that I missed something that should affect my ratings.

---

> ### Author Rebuttal · Authors · 2023-08-28
>
> We appreciate the reviewer's recognition of our dataset as a valuable testbed for assessing the commonsense reasoning abilities of vision-language models. The emphasis on social norms and morals not only validates the intent behind our research but is also encouraging. We are also gratified that you find our data construction and quality control methods sound, and consider our writing to be clear and straightforward. We will address specific concerns and offer further clarification below.
>
> **The coverage of situation types.**
>
> We agree that the coverage/diversity of situations is an important property of the dataset. In Figure 15 (in Appendix), we demonstrated the broad topical coverage within the NormLens dataset. We employed the Berkeley Neural Parser (Kitaev and Klein, 2018) for the illustrative breakdown. Also, to enhance the diversity, we utilized several techniques:
>
> - (L182-L191 in the main paper) We concatenated three datasets for a source — Sherlock (Hessel et al., 2022), Localized Narratives (Pont-Tuset et al., 2020), and COCO Captions (Lin et al., 2014) dataset. Each has unique design principles; Sherlock provides inferences, Localized Narratives offers fine-grained details, and COCO captions present representative captions for the given images. Using these image descriptions as input for ChatGPT, we generate a wide range of situations.
> - (L240-L247 in the main paper) To enhance context diversity, we filtered out scenarios where the generated situation includes commonly produced keywords.
>
> While these efforts may not encompass every possible situation in the world, expanding our framework to address a broader range of situations is a promising future direction we aim to pursue.
>
> **The types of situations/social norms**
>
> For detailed insights into the dataset, we offer visualizations in Figure 5 (Appendix) that showcase the actions and image descriptions within NormLens. For further clarity, we will include illustrative examples from our NormLens dataset along with additional visualizations (*e.g.,* the actual data examples) in the final version of the main paper to better showcase the content of our dataset.
>
> **Precision for NormLens-MA**
>
> We use precision since there can be multiple acceptable answers per sample. To accommodate the subjectivity of moral judgment, we see to it that each sample be annotated by five human annotators. Hence, precision aligns with correctness better than accuracy in case of disagreement among annotators. We will clarify our choice for using precision in the final version.
>
> **Finetuning VLM on NormLens**
>
> Thank you for this valuable suggestion! In response to this suggestion, we additionally run experiments to investigate how the performance of the model (InstructBLIP Flan-13B) varies when fine-tuned on the NormLens dataset. We devised two settings: (1) train the model on NormLens-HA and evaluate it on NormLens-MA, and (2) conversely, train the model on MA and assess it on HA. The results are presented below:
>
> | HA (precision) | Wrong. | Okay. | Impossible. | Avg. |
> |----------------|--------|-------|-------------|------|
> | MA fine-tuning | 18.7   | 94.9  | 20.9        | 44.8 |
> | Four-shot      | 24.6   | 98.6  | 2.5         | 41.9 |
>
> | HA (bleu2)     | Wrong. | Okay. | Impossible. | Avg. |
> |----------------|--------|-------|-------------|------|
> | MA fine-tuning | 6.0    | 34.1  | 11.8        | 17.3 |
> | Four-shot      | 10.8   | 25.7  | 1.2         | 12.5 |
>
> | HA (rouge-l)   | Wrong. | Okay. | Impossible. | Avg. |
> |----------------|--------|-------|-------------|------|
> | MA fine-tuning | 4.5    | 28.1  | 9.4         | 14.0 |
> | Four-shot      | 7.0    | 23.6  | 0.8         | 10.5 |
>
> | HA (meteor)    | Wrong. | Okay. | Impossible. | Avg. |
> |----------------|--------|-------|-------------|------|
> | MA fine-tuning | 4.5    | 23.5  | 7.9         | 12.0 |
> | Four-shot      | 6.5    | 16.9  | 0.7         | 8.0  |
>
> | MA (precision)    | Wrong. or Impossible. | Wrong. or Okay. | Okay. or Impossible. | Avg. |
> |----------------|--------|-------|-------------|------|
> | HA fine-tuning | 22.5   | 93.2  | 99.7        | 71.8 |
> | Four-shot      | 13.7   | 100.0 | 99.2        | 71.0 |
>
>
> Results show that fine-tuning the model on NormLens leads to modest gains in precision (HA Avg. precision: +2.9%, MA Avg. precision: +0.8%) and significant boosts in explanation scores (HA Avg. BLEU-2: +4.8, HA Avg. ROUGE-L: +3.5, HA Avg. METEOR: +4.0) compared to few-shot settings. These results are intuitive, as supervised fine-tuning is commonly beneficial in various ML tasks. But still, fine-tuned InstructBLIP underperforms SM equipped with GPT-4, underscoring the limitations of current VLM models.
>
> **About Image Diversity**
>
> Thank you for bringing us attention to address the diversity of the image! In Figure 15-(b) (in Appendix), we displayed the top 30 nouns that appeared in the image caption, showing the image diversity of our dataset.
>
> In response to the reviewer's question, we ran additional data analysis to show the image diversity:
>
> - We ran DETR (`facebook/detr-resnet-50` from Huggingface) to detect objects that are displayed in images, and found that there are 78 unique objects shown in the images. We will include the exact number in the final version.
>
> - We ran image classification model (`google/vit-base-patch16-224` trained on ImageNet-21k from Huggingface) to classify images, and found that there are 454 unique classes (among 1000 classes) shown in the images.
>
> Moreover, we commit to incorporate visualizations to show the diversity of images within our dataset, drawing inspiration from Liu et al., 2021, particularly “Figure 4”, which illustrates the image feature distribution of the dataset using ResNet embeddings with UMAP.
>
> **References**
>
> - Liu et al., 2021, Visually Grounded Reasoning across Languages and Cultures, In EMNLP
> - Hessel et al., 2022,  The Abduction of Sherlock Holmes: A Dataset for Visual Abductive Reasoning. In ECCV
> - Pont-Tuset et al., 2020, Connecting vision and language with localized narratives. In ECCV
> - Lin et al., 2014, Microsoft coco: Common objects in context. In ECCV
> - Kitaev and Klein, 2018, Constituency parsing with a self-attentive encoder. In ACL

---

### Meta-Review · Area_Chair_i7LZ · 2023-09-18

**Recommendation:** 4

**Metareview:**

This paper introduces a benchmark dataset for visually grounded commonsense reasoning that allows testing models’ predictions coupled with explanations from the model. They compare different model architectures (VLMs/LLMs/combinations of them) and outline a data augmentation/construction pipeline that improves performance of models that train on this new data. All reviewers agree that the benchmark is well constructed and validated and is a good contribution to the community to further vision-language research and evaluate VLN models to a better degree.

---

### Decision · Program_Chairs · 2023-10-07

**Decision:**

Accept-Main

**Comment:**

This paper introduces a benchmark dataset for visually grounded commonsense reasoning that allows testing models’ predictions coupled with explanations from the model. They compare different model architectures (VLMs/LLMs/combinations of them) and outline a data augmentation/construction pipeline that improves performance of models that train on this new data. All reviewers agree that the benchmark is well constructed and validated and is a good contribution to the community to further vision-language research and evaluate VLN models to a better degree.